# Polycomb Alterations in Acute Myeloid Leukaemia: From Structure to Function

**DOI:** 10.3390/cancers15061693

**Published:** 2023-03-09

**Authors:** Teerna Bhattacharyya, Jonathan Bond

**Affiliations:** 1Systems Biology Ireland, School of Medicine, University College Dublin, Belfield, D04V1W8 Dublin 4, Ireland; 2Children’s Health Ireland at Crumlin, Crumlin, D12N512 Dublin 12, Ireland

**Keywords:** epigenetics, polycomb, acute myeloid leukaemia, protein structure, genetic interactions

## Abstract

**Simple Summary:**

Epigenetic factors control how genes are expressed in different cell types. Members of the Polycomb Repressive Complexes (PRCs) are critical for epigenetic control of gene transcription in blood cells, and mutations and deletions in these factors are common in the blood cancer acute myeloid leukaemia (AML). This review article provides an overview of how the structure and function of PRCs are affected by genetic alterations in AML, with a primary focus on PRC2 core factors. We document how mutations and deletions in PRC2 factors are linked to other AML-associated genetic alterations and discuss how these observations might inform potential treatment avenues in future.

**Abstract:**

Epigenetic dysregulation is a hallmark of many haematological malignancies and is very frequent in acute myeloid leukaemia (AML). A cardinal example is the altered activity of the Polycomb Repressive Complex 2 (PRC2) due to somatic mutations and deletions in genes encoding PRC2 core factors that are necessary for correct complex assembly. These genetic alterations typically lead to reduced histone methyltransferase activity that, in turn, has been strongly linked to poor prognosis and chemoresistance. In this review, we provide an overview of genetic alterations of PRC components in AML, with particular reference to structural and functional features of PRC2 factors. We further review genetic interactions between these alterations and other AML-associated mutations in both adult and paediatric leukaemias. Finally, we discuss reported prognostic links between PRC2 mutations and deletions and disease outcomes and potential implications for therapy.

## 1. Polycomb Repressive Complexes

The Polycomb Group (PcG) of proteins are key regulators of metazoan development that were first identified in *Drosophila* genetic screens. The name Polycomb derives from the anatomical changes in *Drosophila* lacking these genes, wherein the comb-like set of bristles on the posterior legs of the male flies undergo a homeotic transformation, now known to be due to loss of Polycomb-mediated epigenetic repression of *HOX* (homeobox) gene transcription [1]. Genetic and phenotypic activities of *PcG* genes in *Drosophila* are now very well characterised, facilitated by the relatively simple one gene-one function correlation in this setting [2,3].

In mammals, Polycomb proteins form large multimeric complexes to exert their function as epigenetic regulators of gene expression. Among these complexes, Polycomb Repressive Complex 1 and 2 (PRC1 and 2) have been the best described so far, with other existing variants having less thorough functional characterisation. Canonical forms of PRC1 and 2 target histones 2A and 3 within the core nucleosome and work cooperatively toward chromatin compaction and transcriptional repression. These activities are believed to be mainly mediated through the enzymatic activity of each complex, with PRC1 targeting histone H2A to produce monoubiquitylated lysine 119 (H2AK119) and PRC2 methylating histone 3 at lysine 27 to produce mono-, di- and tri-methylated H3K27 (H3K27me1, H3K27me2 and H3K27me3).

At the catalytic core of PRC1, RING (Really Interesting New Gene) and PCGF (Polycomb Group Ring Finger 1) proteins form a heterodimer and associate with CBX (Chromobox), PHC (Polyhomeotic Homolog) and SCMH (Scm homolog) molecules to form the canonical PRC1 complex, whereas several accessory proteins are involved in the non-canonical (nc) complexes and guide the genomic localisation of the PRC1 machinery depending on the cellular context [4]. The canonical PRC2 complex consists of EZH1/2 (Enhancer Of Zeste 1/2), Suppressor Of Zeste 12 (SUZ12) and Embryonic Ectoderm Development protein (EED) at its catalytic core along with Retinoblastoma-Binding Protein 4/7 (RBBP4/7) (Figure 1). PRC2 has fewer accessory proteins documented so far in comparison with PRC1; EPOP (Elongin BC And Polycomb Repressive Complex 2-Associated Protein), ELOB (Elongin B), ELOC (Elongin C), LCOR/LCORL or PALI1/2 (Ligand Dependent Nuclear Receptor Corepressor/Ligand Dependent Nuclear Receptor Corepressor-Like) and PCL1-3 (Polycomb-Like 1–3) form the PRC2.1 non-canonical complex while JARID2 (Jumonji And AT-Rich Interaction Domain Containing 2) and AEBP2 (Adipocyte Enhancer-Binding Protein 2) are part of the PRC2.2 complex (Figure 1). The methyltransferase activity of EZH1/2 is regulated by the other core PRC2 proteins, while the accessory proteins modulate other functions, including DNA binding at CpG islands [5,6,7]. EZH1/2 is incapable of catalysis alone and only attains an active conformation that allows mono-, di- or tri-methylation upon assembly of the PRC2 complex. The overall sequence identity between EZH1 and EZH2 is below 70%, but key functional sites such as the SET domain are well conserved between the paralogs. While EZH2 is more abundant and catalytically active than EZH1, the latter can partially substitute the function of EZH2 and may even heterodimerise to target PRC2 dimers to specific chromatin sites [8,9,10].

In line with the original key roles for PcG factors in *Drosophila* development, Polycomb proteins have been shown to play critical roles in many aspects of mammalian differentiation, including haematopoiesis [1,9,11]. Much of our knowledge in this field has been provided by murine experimental models of deletion and depletion of PRC1 and PRC2 components that revealed significant roles for these complexes in blood cell development. For example, while overexpression of the ncPRC1.1 factor *BCOR* (BCL-6 Interacting Corepressor) leads to repression of *HOXA* genes, depletion of core *PCGF1* (PRC1 factor Polycomb Group Ring Finger 1) and accessory factor *KDM2B* (Lysine Demethylase 2B) in haematopoietic stem and progenitor cells (HSPCs) leads to impaired myeloid proliferation [12,13,14].

Several studies have identified key roles for PRC2 factors in normal haematopoiesis. Ezh2 has been reported to modulate haematopoietic stem cell (HSC) differentiation and senescence and to prevent stem cell exhaustion [15], and to regulate the HSC cell cycle [16]. Murine knockout of PRC2 core factor *Suz12* was shown to lead to significant loss of long-term HSCs and myeloid progenitors, along with impaired erythropoiesis. In-depth analyses of these results revealed the dispensability of *Suz12* in some haematopoietic lineages and the possibility of an *Ezh1*/*Ezh2* independent function of Suz12 during blood cell development [17]. Similarly, it was reported that *Ezh2* is crucial for lymphocyte division and cell cycle regulation during lymphopoiesis [18,19]. Conditional knockouts in mice showed that haploinsufficiency of *Eed* induced abnormal function and differentiation of HSCs leading to leukaemogenesis and that Eed depletion led to impaired fetal haematopoiesis in mice [20,21]. Overall, the PRC2 complex is considered to play a pivotal role in the transition from proliferative fetal HSPCs to quiescent adult HSCs in a differentiation stage-specific and dose-dependent manner [16,20].

Given these key roles in normal blood cell development, it is unsurprising that acute leukaemias frequently harbour somatic mutations or deletions of genes coding for PRC components. While genetic alterations in PRC1 factors are rare, reduced function of PRC2 due to mutation or deletion is common in acute myeloid leukaemia (AML) and the immature subgroup of T-acute lymphoblastic leukaemia (T-ALL), as well as the rarer T/myeloid mixed phenotype acute leukaemia (MPAL) subtype. While PRC2 alterations are known to be linked to poor outcomes and resistance to existing chemotherapies in both AML and T-ALL [22,23,24,25,26], the precise molecular mechanisms of these effects are not fully understood. In this review, we explore the recent findings on the molecular effects of epigenetic disruption linked to PRC2 alterations in AML, with reference to structural features of the PRC2 complex, and highlight the lacunae in this field.

## 2. Structural and Functional Regions of the PRC2 Complex

PRC2 core proteins contain several domains that are directly or indirectly involved in the assembly of the complex and regulation of catalytic activity. The activities of these domains are best considered holistically in the context of the multimeric assembly rather than by how they relate to individual protein functions. In the following sections, we discuss the structural aspects of these key domains and their functional contribution to PRC2 methyltransferase activity.

### 2.1. Catalytic and Regulatory Regions of PRC2

The first crystal structure of murine Ezh2 bound to Eed (corresponding to constructs shorter than the full-length proteins) was published in 2007 [27]. The first structure of the human holo-PRC2 complex was determined using low-resolution electron microscopy (EM) in 2012 [28]. Since then, numerous structures of improved resolution corresponding to either the core assembly or accessory protein-bound PRC2 have been solved. Broadly, the PRC2 assembly comprises catalytic and regulatory lobes that are directly involved in methyltransferase activity, while protein-protein interaction lobes mediate PRC2 intra-complex subunit interactions driving both canonical and non-canonical functions of the complex (Figure 2A).

The catalytic lobe of PRC2 is made up of multiple domains or motifs from two core proteins: the SET domain (**S**wi3, Ada2, N-Cor, and TFIIIB or SANT domain, Su(var)3-9, **E**nhancer-of-zeste and **T**rithorax domain) and CXC (Cysteine-rich) domain of EZH1/2, and the VEFS motif of SUZ12. The enzymatic function is provided by the EZH2 SET domain, which is a common feature of human methyltransferases. Within this domain, several residues stabilise the long aliphatic tail of H3K27 in an aromatic pocket, while additional residues stabilise the S-adenosyl methionine or SAM cofactor, thereby helping the catalytic lobe to approximate to the target K27 residue (Figure 2A,B). In the absence of full PRC2 complex assembly, the post-SET domain C-terminal tail of EZH2 folds and blocks the lysine-binding site to prevent substrate binding and methyltransferase activity. Only upon assembly of the core proteins does this auto-inhibited conformation change to the active conformation required for catalysis.

Several regions of the regulatory lobe play key roles in modifying PRC2 enzymatic activity. This includes a domain involving SUZ12 and EZH2 that promotes the catalytic activity of the PRC2 complex. The key region in EZH2 in this interaction is the GWG motif within the SET domain that contains a well-conserved tryptophan residue (W) (Figure 2B) which is crucial for catalytic function. Approximation of this GWG motif and the SET activation loop of EZH2 is mediated by extensive hydrophobic contacts with the SUZ12 VEFS region that maintain the catalytic site in an active conformation (Figure 2B).

A further domain constituted by EED and EZH2 plays an important role in fine-tuning the catalytic activity of the complex. This interaction involves a long α-helix in EZH2 termed the EED-binding motif domain (EBD) and the β-addition motif or BAM from the N-terminal region of EZH2, which lasso around EED’s WD-repeat seven-bladed β-propeller domain (Figure 2C). This conformation facilitates interaction between the stimulation-responsive motif (SRM) and SET-activation loop (SAL) of EZH2 and EED that renders the PRC2 complex highly sensitive to the conformational changes induced by the binding of EED to H3K27me3, thereby providing a positive feedback loop for the EZH1/2 activity.

### 2.2. Interaction Regions of PRC2

The protein key to the docking region is SUZ12, a core PRC2 factor that interacts with RBBP4/7 as well as accessory factors of the complex (AEBP2, PHF19 or PCL3, JARID2, etc.). Like EED, RBBP4/7 also possesses a WD propeller domain with repeating β-sheets which interacts closely with N-terminal residues of SUZ12 from the WD-binding or WDB1 and 2 domains. The C2 and Zinc finger binding (ZnB) domains of SUZ12 act as platforms for pleiotropic protein-protein interactions between core PRC2 members and accessory proteins (Figure 2A). Accessory factors AEBP2 and PHF19 or PCL3 bind via the C2 domain in a mutually exclusive manner, while JARID2, EPOP, and PCL proteins interact with the Zn finger and ZnB domains to form the non-canonical PRC2 complexes PRC2.1 and PRC2.2 (Figure 1). Recruitment of a diverse set of accessory proteins via the docking region endows non-canonical PRC2 complexes with the ability to function according to local epigenetic context, albeit there is considerable overlap among the targets of the non-canonical PRC2 complex.

Apart from the catalytic, regulatory, and docking regions that are part of the molecular machinery of PRC2, additional protein regions dictate how PRC2 accesses active chromatin. These interactions, and the catalytic activity of PRC2, are further regulated by the chemical state of other histone amino acid residues. For example, structural, biochemical, and genetic analysis of nucleosome-bound PRC2-PCL1 revealed that binding of unmodified (i.e., unmethylated) H3K36 allosterically activates the H3K27 methylation activity of PRC2 [29,30]. In contrast, in the case of di-/trimethylated H3K36 bound PRC2, the interaction between H3K27 and the EZH2 active site is chemically and geometrically disrupted, and as a result, methyltransferase activity is inhibited (Figure 2D) [31]. It has also been shown that PHF19 or PCL3 can each transiently bind to H3K36me3 to stimulate PRC2 methyltransferase activity [32]. PRC2 complex activity is further inhibited by H3K4me3, and it was recently demonstrated that recruitment of JARID2 and AEBP2 to the PRC2 complex could alleviate the inhibition of methyltransferase activity by both H3K4me3 and H3K36me3 to enable transcriptional repression [31].

## 3. PRC2 Alterations in Acute Myeloid Leukaemia

With the advent of high-throughput DNA sequencing, mutations and deletions of genes coding for PRC2 factors came to be increasingly identified in AML [24,33,34,35,36]. In this section, we highlight the alterations in functional domains of PRC2 core proteins that are seen in paediatric and adult leukaemias.

Of note, while the overall mutational genotype of childhood and adult leukaemias differ significantly, the repertoire of PRC2 mutations is similar in each case and further largely overlaps with alterations seen in both T-ALL and MPAL [37,38,39]. In the following sections, we discuss how these alterations map to key PRC2 functional domains. Further, we will highlight differences in AML mutational patterns in adult and paediatric cases while discussing key co-occurring mutations and mutual exclusivities with genes coding for PRC2 components.

### 3.1. Missense Mutations of PRC2 Components in AML

Consortia-led cancer genomics efforts in both paediatric and adult leukaemias have yielded rich datasets of genetic alterations that provide a thorough catalogue of the range of PRC2 alterations found in AML. In Figure 3, we show a map of missense mutations in core PRC2 components from paediatric and adult leukaemia cohorts, comprising the COSMIC [40], TARGET and PCGP datasets within the PeCan Data Portal [41] and the ELAM02 dataset [24,36], which is discussed in the next section. A full list of these variants and their predicted functional consequence is provided in Table 1, along with details of whether these alterations were detected in adult or paediatric AML cohorts.

Out of 34 *EZH2* mutations in AML, 27 map to well-conserved residues belonging to the CXC, SET and post-SET regions that play an important role in H3K27 methylation and transcriptional repression (Figure 3A,B). Two residues, Y641 and R685, that are highly conserved and crucial for stabilisation of the H3K27 side chain and the cofactor (Figure 2B) are mutated in AML, with R685 being a mutational hotspot (R685H and R685C variants are present in respectively 12 and 2 samples out of 54; Figure 3A). Y641 is frequently mutated in B-cell lymphomas, and most of these variants are gain-of-function [42]. However, the EZH2 Y641C mutation found in the AML cell line SKM-1 has been reported to significantly abrogate the histone methyltransferase activity [43,44]. Apart from these alterations, several AML missense mutations map to residues that make polar contacts with the SAM cofactor and H3K27 in the crystal structure (PDB ID: 6WKR, Figure 2A,B), including S664, N688, H689, S690 and E740, which is likely to have a significant effect on the stabilisation of the substrate and cofactor in the catalytic cage. Other mutations at V621, V674 and R697 correspond to positions that are proximal to residues important for the structural stability of the catalytic cavity (Figure 2B). Of note, SIFT and PolyPhen-2 scores for these mutations also predict compromised function (Table 1). Additional mutations are found in domains implicated in other EZH2 functions, including protein-protein interaction: for example, the SANT domain. Regardless of location, all these mutations are predicted to result in decreased EZH2 function (Table 1) [38].

*SUZ12* mutations in AML are found in the ZnB and C2 domains that are important for the recruitment of core and accessory proteins in PRC2 complexes (Figure 3C). In *EED* and *RBBP4*, AML mutations from the COSMIC, PCGP and ELAM02 studies map to the β-propeller constituted WD-binding domains that are important for the stabilisation of the PRC2 complex (Figure 3D,E). The R103Q mutation in the ZnB domain falls at the interface where SUZ12 binds both RBBP4/7 and AEBP2 (as seen in the crystal structure 6WKR). This mutation also has SIFT and PolyPhen-2 scores indicative of disruption of function. The W643R mutant of EED maps to a residue that may interact with JARID2, as captured in the crystal structure 6WKR (Figure 2A, green box). Although the other EED mutant, L196Q, does not fall at any interaction region, the introduction of a polar amino acid (Gln) upon mutation may destabilise the structural repeats of WD-binding domains. The RBBP4 mutant E330K does not appear to significantly destabilise the structure or function of this protein (Table 1).

In Table 2, we list the alterations in PRC2 core and accessory factors found in AML cell lines that are used to experimentally model this disease. While most of these mutations are missense point mutations, a few synonymous and frameshift mutations in PRC2 factor genes are also reported. Of note, SIFT and PolyPhen-2 analyses show that most of the mutations are predicted to be deleterious for the protein and, therefore, complex function. Overall, the repertoire of mutations from Table 1 and Table 2 suggest that loss of PRC2 function is a molecular hallmark of leukaemias that harbour these alterations. However, biological validation of PRC2 loss of function and/or epigenetic dysregulation in these cell lines is currently lacking in most cases.

### 3.2. PRC2 Genetic Interactions in Childhood and Adult AML

The molecular signatures of AML cells vary markedly with patient age, meaning that patterns of mutational co-occurrence observed in adult cases may not necessarily be relevant in paediatric AML and vice versa. In this section, we discuss patterns in genetic interactions of PRC2 in adult and childhood AML.

A seminal study in 2016 reported the genomic classification of adults (18–84 years of age) undergoing treatment for AML [34]. Several driver and co-occurring mutations were reported to involve genes encoding proteins of diverse biological function, including PRC2 components, providing insights into novel gene-gene interactions. Notably, *EZH2* mutations were enriched in cases harbouring the *DEK-NUP214* fusion arising from the translocation t(6;9) and in cases with mutations in *NPM1* (nucleophosmin 1), *CEBPA^biallelic^* (biallelic CCAAT enhancer binding protein alpha), or *TP53* (Tumor protein P53). Of note in this study, other epigenetic regulators had different patterns of genetic interaction. For example, mutations in *IDH2* (Isocitrate Dehydrogenase 2), specifically *IDH2*^R140^ and *IDH2*^R172^, were found to occur with *FLT3* (FMS-like tyrosine kinase receptor-3) alterations, including both *FLT3* internal tandem duplication (FLT3-ITD) and *FLT3* tyrosine kinase domain (FLT3-TKD) mutations. Another study in the same year focused on core binding factor AML (CBF-AML) in patients aged 1–60 years. Although rare, CBF-AML was reported to harbour *EZH2* mutations in 3% of this cohort, and these alterations were enriched in cases that relapsed [48].

In 2018, whole-exome sequencing and genomic analyses of tumour samples from 562 patients from a predominantly older patient cohort were reported as part of the Beat AML Master Trial [35], further enriching the database of PRC2 alterations in adult AML. Eleven genes that had not been reported to be mutated in AML in previous studies were reported as new alterations in this cohort. Analysis of co-occurrence and exclusivities of frequently mutated genes revealed interesting patterns of genetic interactions that might inform the understanding of molecular mechanisms in these diseases. For example, *EZH2* mutations were found to be mutually exclusive with both *FLT3-ITD*s and *SRSF2* (serine- and arginine-rich splicing factor) mutations (FDR-corrected statistical significance were *p* < 0.05 and *p* < 0.1, respectively).

A landmark analysis of the molecular landscape of AML in younger patients was provided by Bolouri, Meshinchi and co-workers, who performed a comprehensive genomic analysis of nearly 1000 childhood, adolescent and young adult AML patients aged 8 days to 29 years who were part of the Children’s Oncology Group (COG)—Therapeutically Applicable Research to Generate Effective Treatments (TARGET) AML initiative highlighted differences in mutational patterns [33].

This study highlighted several major differences in the mutational spectrum observed in adult and paediatric AML. In particular, recurrent structural alterations, fusions and focal copy number aberrations were demonstrated to be much more common signatures of childhood AML than adults, while mutations in *TP53* and epigenetic regulator *DNMT3A* (DNA Methyltransferase 3 Alpha) that are relatively common in adults are rarely, if ever, found in childhood AML cases. In this TARGET study, several paediatric-specific *FLT3* mutations were detected, while hotspots of *MYC* (MYC Proto-Oncogene, BHLH Transcription Factor) alterations exclusive to childhood AML were also reported. Further, *WT1 (*Wilms Tumor 1)*, KIT* (KIT Proto-Oncogene, Receptor Tyrosine Kinase)*, CBL (*Cbl Proto-Oncogene)*, GATA2 (*GATA Binding Protein 2)*, SETD2 (*SET Domain Containing 2, Histone Lysine Methyltransferase) and *RAS* pathway associated genes were mutated more commonly in this younger cohort than in adult cases.

Given these mutational differences, it is logical that genetic interactions of *EZH2* in paediatric and adult AML should also be different. In the TARGET study, unlike in previous adult cohorts, *WT1* mutations were found to be mutually exclusive with mutations of either *EZH2* or the Polycomb factor *ASXL1* (Additional Sex Combs Like 1 protein). Of note, WT1 is known to be involved in the recruitment of EZH2 to chromatin, suggesting that this genetic interaction might be underpinned by a functional dependency and/or synthetic lethality in this case.

A study that was published in the same year as TARGET identified mutations and deletions of PRC2 factors in 32/220 patients (14.5%) from the French ELAM02 paediatric AML study [24,36]. In many cases, PRC2 loss was linked to deletions of varying extent on the portion of chromosome 7q on which *EZH2* is located. This analysis reported a relatively high incidence (26%) of PRC2 mutation or deletion in the standard risk subgroup, which comprised cases with translocations in core binding factor (CBF) core components (i.e., *RUNX1-RUNX1T1* or *CBFβ-MYH11*-expressing cases). Of note, wild-type (WT) CBF has been shown to recruit PRC1 to chromatin in a PRC2-independent manner in haematopoietic cells, and it has been subsequently shown that this activity is subverted in AMLs that express *CBFβ-MYH11* [49,50].

In Figure 4, we provide a visual summary of genetic interactions in a number of adult and paediatric AML cohorts. *EZH2* accounts for 2.4% of the total number of mutations analysed for adult AML in the OncoPrint [51] (datasets included are from Beat AML [35] and The Cancer Genome Atlas (TCGA) Research Network (2013–2018) [52,53]), while mutations of other core factors *SUZ12, EED* and *RBBP4* contribute 1.5, 0.7 and 0.5% to the data, respectively. Among non-canonical factors, *JARID2* and *AEBP2* account for 0.2% each and *PHF1* or *PCL1*, *PHF19* or *PCL3* and *LCOR* or *PALI1* correspond to 0.1, 0.1 and 0.3% of the mutations, respectively. *PCL1, PCL3* and PALI1 harbour both missense and splice mutations, while *JARID2* and *AEBP2* harbour splice, missense, and truncating mutations. For paediatric AML (corresponding to the TARGET AML study [33]), alterations are limited to *EZH2* due to the limited number of studies. A total of 1.3% of the total number of alterations in the paediatric AML dataset shown in Figure 4 are contributed by *EZH2,* all of which are missense and truncating mutations.

## 4. Therapeutic Implications of PRC2 Alterations in AML

Mutations and deletions in genes encoding core PRC2 factors have been documented in studies across multiple AML cohorts, with *EZH2* alterations being the most common alteration (Figure 4). In the following sections, we discuss the available data that link PRC2 alterations to AML prognosis and potential implications for therapy.

### 4.1. Prognostic Associations of PRC2 Alterations in AML

Several reports have identified PRC2 mutations and deletions as predictors of unfavourable outcomes in AML. A landmark genomics study in 2016 revealed mutations in genes encoding chromatin, splicing, and transcriptional regulators, including *EZH2*, as poor prognostic markers in adult AML [34]. An analysis of a separate cohort of 124 adult patients with AML found that low levels of EZH2 expression, which corresponded with reduced levels of H3K27me3, correlated strongly with poor overall and event-free survival. Of note, not all cases in this study were found to have genetic alterations in *EZH2* or other PRC2 core factors, suggesting that reduced PRC2 function might be a convergent molecular mechanism for AML treatment resistance. Low expression of EZH2 and H3K27me3 was also seen in about half of all samples of relapsed leukaemia in this cohort [25].

In the case of paediatric AML, a study that used targeted sequencing of Polycomb genes and copy number assessment by single nucleotide polymorphism (SNP) array of 222 AML samples from the French cohort ELAM02 revealed strong associations between alterations of genes encoding PRC2 factors and poor prognosis [24,36]. In this study, PRC2 core components (*EZH2*, *SUZ12*, and *EED*) were altered in 14.5% of the cases, with mutations including missense, frameshift, and in-frame insertions. The relative frequencies of mutations/deletions in this cohort were as follows: *EZH2* 1.4% mutations/10% deletions, *SUZ12* 2.3% mutations/0.9% deletions and *EED* 0.5% mutations/1.4% deletions. While these alterations did not correlate with final outcomes in the standard risk treatment subgroup in this study, cases with intermediate and high cytogenetic risk and PRC2 mutation or deletion had significantly lower 5-year overall survival (40.9% mutated/deleted vs 69.1% non-mutated/deleted), event-free survival (31.8% mutated/deleted vs 50.8% non-mutated/deleted) and markedly increased rates of relapse (57.3% mutated/deleted vs 38.0% non-mutated/deleted), strongly suggesting that PRC2 haploinsufficiency is a marker of biological aggressiveness in these leukaemias.

Of note, PRC2 inactivation has also been shown to be linked to treatment resistance in T-ALL, which shares many similarities in mutational genotype with AML, especially among cases with an immature/early thymic precursor (ETP)-ALL phenotype. Chemoresistance in these cases has been shown to be linked to reduced mitochondrial priming caused by indirect up-regulation of the mitochondrial chaperone *TRAP1* (TNF Receptor Associated Protein 1) [26], a hypothesis that has yet to be tested in AML. In both T-ALL and AML, the broad actions of PRC2 across the leukaemia genome means that the exact molecular mechanisms of therapy resistance in leukaemia are likely to be multifactorial and require much further research.

### 4.2. Epigenetic Therapeutic Avenues towards Improved Treatments for AML

AML is an important example of epigenetic disruption in cancers, and epitherapies are a promising avenue for AMLs with resistance toward conventional therapies [22,54,55]. BET protein inhibitors were recently identified as targetable vulnerabilities for PRC2-depleted T-ALL in adults [56]. Given the similar repertoire of PRC2 alterations in T-ALL and AML, this suggests that this strategy might be adopted in AML, albeit this approach has yet to be validated at the time of writing.

Some recent reports have documented the disappointing efficacy of epigenetic target therapies when used as single agents, stimulating recent efforts to incorporate these treatments in drug combination approaches [57,58]. Rational pathways to design these combination strategies should ideally be based on knowledge of the downstream molecular effects of PRC2 mutations and deletions in AML or by pharmaceutical manipulation of functional pathways that govern epigenetic activity. As an example, multi-drug resistance in AML was reported to correlate with low levels of EZH2 expression due to Cyclin Dependent Kinase 1 (CDK1) and Heat Shock Protein 90 (HSP90) mediated proteasomal degradation. This pathway could be targeted by proteasomal inhibition with bortezomib, which restored EZH2 levels in multi-drug resistant AML cells and led to corresponding reductions in *HOX* gene expression in this study, promising potential approaches to overcome treatment resistance [25].

An intriguing recent report has also identified EZH2 inhibition as a potential strategy to overcome treatment resistance [59]. This effect relies on EZH2 inhibition causing decondensation of H3K37me3-marked heterochromatin, thereby enhancing chromatin accessibility to DNA-damaging cytotoxic agents such as doxorubicin. This approach was shown to enable the delivery of lower doses of chemotherapeutic agents in both cell lines and preclinical models in this study.

Separately, EZH2 inhibitors combined with glucocorticoids or combination chemotherapy were reported as a promising treatment strategy for relapsed ALL with *NSD2* mutations [60], providing further evidence that manipulation of PRC2 function might be therapeutically beneficial in some leukaemia subtypes. It is to be hoped that the synergistic effects of epigenetic drugs targeting PRC2 in combination with inhibitors of other regulatory proteins or with conventional cytotoxic agents may improve treatment outcomes of AML and reduce dependence on non-specific chemotherapies that can result in significant long-term toxicities in childhood AML in particular.

## 5. Conclusions

PRC2 genetic alterations are frequent in paediatric and adult AML, directly linking epigenetic dysregulation to oncogenesis in these leukaemias. The protein structural locations of PRC2 mutations are similar in childhood and adult AML, but the associated effects on epigenetic function are likely to be affected by age-dependent variability in other leukaemia-associated mutations.

Several studies have now linked reduced PRC2 function to poor prognosis and resistance to routine chemotherapies. Much work remains to be done to elucidate the precise molecular mechanisms by which these mutations affect disease biology and how alterations in other functional pathways might further alter epigenetic activity and/or responses to treatment. It is to be hoped that improved knowledge in this area might provide rational avenues to design better therapies to treat PRC2-altered AML.

## Figures and Tables

**Figure 1 cancers-15-01693-f001:**
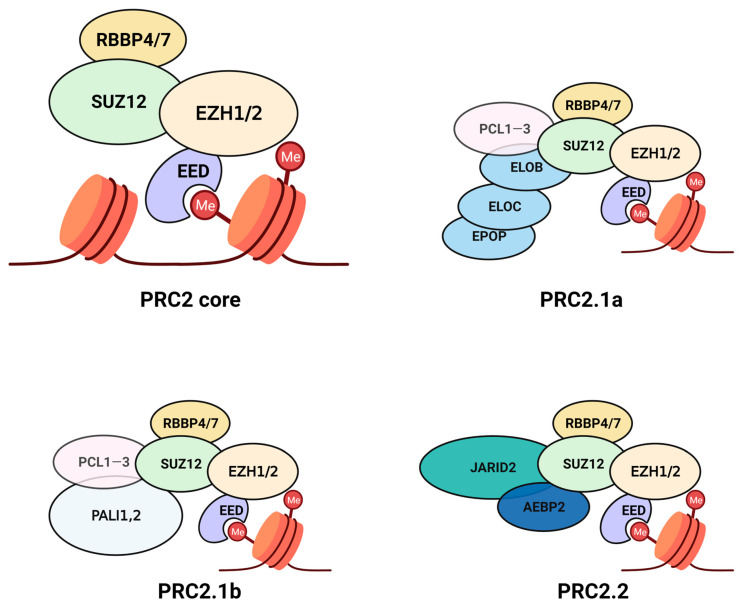
The Polycomb repressive complexes targeting histone 3 consist of a catalytic core and multiple accessory proteins that vary by function and cellular context. Complex subunits are depicted in proportion to their size (i.e., number of amino acid residues). Polycomb repressive complex 2 (PRC2) consists of a tetrameric core, where methyltransferase activity of EZH1/2 is stimulated by EED binding to methylated H3K27, while SUZ12 and RBBP4 stabilise the overall complex. The non-canonical or ncPRC2 complexes are formed by the recruitment of accessory proteins by SUZ12. EPOP, ELOC, ELOB, and PCL1-3 are recruited in ncPRC2.1a and PALI1—2 and PCL1—3 in ncPRC2.1b complexes, while JARID2 and AEBP2 are accessory components of the ncPRC2.2 complex. This figure was created with BioRender.com.

**Figure 2 cancers-15-01693-f002:**
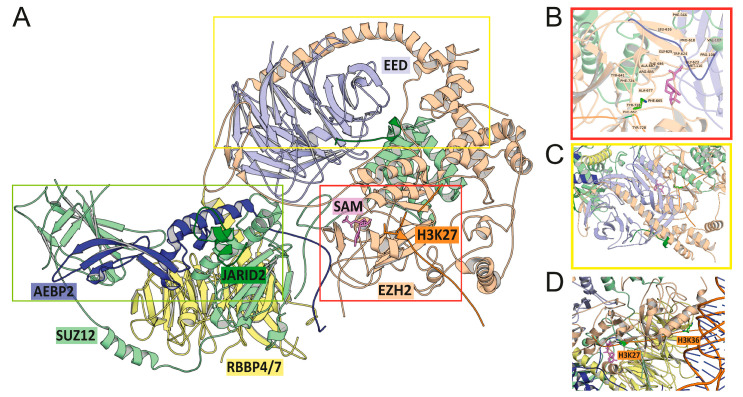
Crystal structure of PRC2 (PDB ID: 6WKR [31]) showing functionally important regions. (**A**) EZH2 (beige) engages with SUZ12 (pale green), H3K27 peptide (orange) and SAM cofactor (pink) in the catalytic domain (red box) and with EED (cornflower blue) in the regulatory domain (yellow box). SUZ12 is bound to RBBP4 (pale yellow), AEBP2 (deep blue) and JARID2 (dark green) in the interaction domain (green box) that is distant from the catalytic and regulatory domains. (**B**) The histone methyltransferase site of PRC2. The cofactor SAM and long aliphatic side chain of H3K27 are stabilised by hydrophobic and aromatic amino acids from the SET domain of EZH2 (H3K37 side chain: sticks; PRC2 residues: labelled). (**C**) The regulatory domain of PRC2. EED binding domain or EBD is a long α-helix of EZH2 that wraps around the β-propeller domain of EED, forming a conformation-sensitive feedback loop for EZH1/2 activity. (**D**) Close-up of PRC2 bound to histone 3, with the two residues that are methylated on H3, i.e., H3K27 and H3K36, highlighted in green (side chains shown as sticks). PRC2 core members EED and SUZ12 are also shown alongside the SAM cofactor. Histone associated with DNA is shown on the lower right side of the panel. This figure was created using the PyMOL Molecular Graphics System v.2.5.4, Schrödinger, LLC.

**Figure 3 cancers-15-01693-f003:**
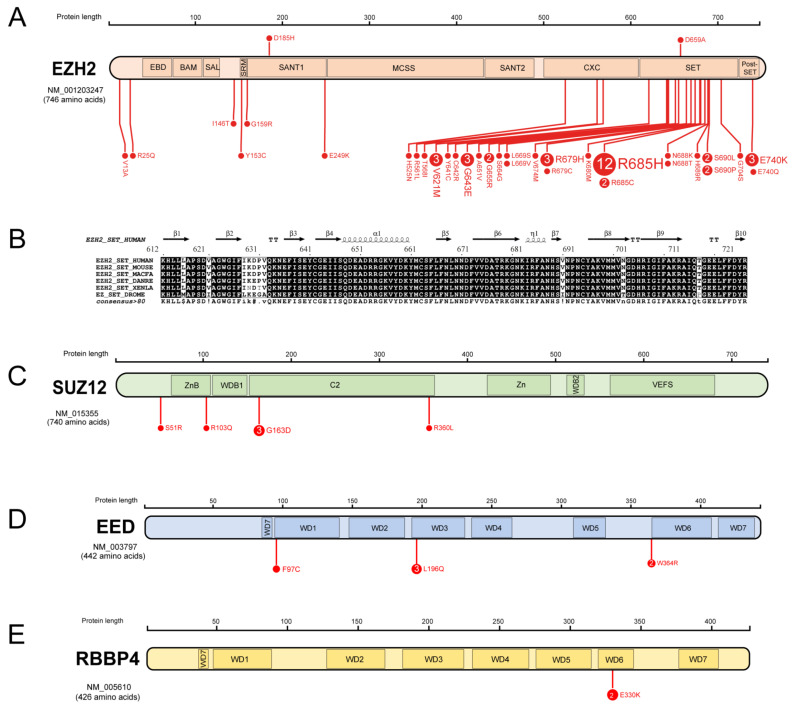
Missense mutations seen in AML mapped to core PRC2 components. (**A**) *EZH2* mutations span the entire coding sequence but are more frequent in the CXC and SET domains that are important for catalysis. (**B**) SET domain sequences from different species reflecting the conservation of residues around AML mutational hotspots. (**C**) Zinc finger binding or ZnB and the C2 domains harbour *SUZ12* mutations that may affect the inter and intra-complex interactions. (**D**,**E**) show the mutations on WD repeats of EED and RBBP4, respectively. Images (**A**,**C**–**E**) have been adapted from PeCan ProteinPaint [41], and (**B**) was created using ESPript—https://espript.ibcp.fr on 24 October 2022 [45].

**Figure 4 cancers-15-01693-f004:**
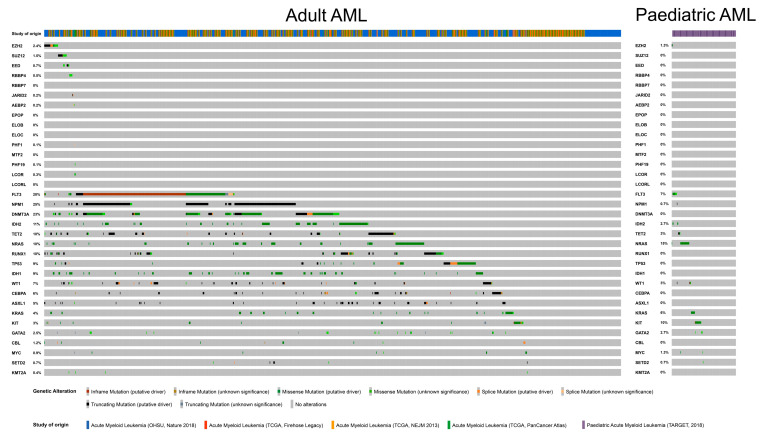
OncoPrint of genetic interactions involving mutations found in adult and paediatric patients with AML. The genes that are frequently mutated in adult and paediatric AML were queried in the Beat AML [35] (study of origin marked as OHSU, Nature 2018), TCGA [52,53] and TARGET 2018 [33] datasets. The genes corresponding to PRC2 factors are listed first, followed by other frequently mutated genes in order of relative frequencies of mutations in adult AML. For visualisation purposes, genes coding for PRC2 factors are shown at the top of these lists, with other AML-associated mutations depicted in descending order of occurrence in adult AML samples. The individual oncoprints with higher image resolution can be found at- https://tinyurl.com/yysensxu (adult AML) and https://tinyurl.com/y7s7xdfa (paediatric AML) and were obtained from OncoPrint within cBioPortal [51] on 19th January 2023.

**Table 1 cancers-15-01693-t001:** Missense mutations in PRC2 core components in AML studies. Information was extracted from COSMIC, TARGET and PCGP datasets within the PeCan Data Portal [41] and the ELAM02 dataset [24,36] and whether the mutation was detected in a paediatric or adult cohort or in both is indicated in the fourth column in the table. Predictions of functional consequences by the SIFT [46] (higher score = more impaired function) and PolyPhen-2 [47] (lower score = more impaired function) algorithms are provided for these mutations. Functionally important regions of the core PRC2 proteins- CXC and SET (of EZH2), VEFS (of SUZ12) and WD domains of EED and RBBP4 harbour multiple mutations and include hotspot residues of typically varying allele frequencies.

PRC2 Component	Domain	Mutation	Paediatric/Adult	SIFT	PolyPhen-2
EZH2		V13A	Adult	0.52	0
	SBD	R25Q	Paediatric	0	0.99
		I146T	Adult	0	0.43
	SRM	Y153C	Adult	0	1
	SANT1 domain	G159R	Adult	0	1
		D185H	Paediatric	0	0.41
		E249K	Adult	0	0.82
	CXC domain	H525N	Adult	0	1
		R561L	Adult	0	1
		T568I	Adult	0	0.03
	SET domain	V621M	Adult	0.02	1
		C642R	Adult	0	1
		G643E	Both	0	1
		A651V	Adult	0.03	0.61
		G655R	Both	0.01	1
		D659A	Adult	0.02	1
		S664G	Adult	0.01	1
		L669S	Adult	0.03	1
		L669V	Adult	0.03	1
		V674M	Adult	0	1
		R679H	Both	0	0.65
		R679C	Adult	0	1
		K680M	Both	0.04	1
		R685H	Both	0	1
		R685C	Both	0	1
		N688K	Adult	0	1
		N688T	Adult	0	1
		H689R	Adult	0	1
		S690L	Both	0	1
		S690P	Adult	0	1
		G704S	Adult	0.04	1
	post-SET domain	E740K	Both	0.03	0.12
		E740Q	Paediatric	0.03	0.91
SUZ12		S51R	Adult	0	0.38
	ZnB domain	R103Q	Adult	0	1
	C2	G163D	Adult	0.08	0.98
		R360L	Paediatric	0.11	0
EED	WD1 repeat	F97C	Paediatric	0.01	1
	WD3 repeat	L196Q	Adult	0.01	1
	WD3 repeat	W364R	Paediatric	0	1
RBBP4	WD6 repeat	E330K	Adult	0.05	0.84

**Table 2 cancers-15-01693-t002:** PRC2 mutations in AML cell lines. Information was extracted from the DepMap portal (https://depmap.org/portal, accessed on 6th July 2022). The structural domain of the corresponding protein is indicated along with the SIFT [46] and PolyPhen-2 [47] scores to indicate the predicted functional consequence. A low SIFT score and a high PolyPhen-2 score indicates a damaging mutation. N/A: Prediction of functional consequence was not possible for some insertion and deletion events.

PRC2 Component	AML Cell Line	Type of Mutation	Protein Domain	Mutation	SIFT	PolyPhen-2
EZH2	PL21	SNV	CXC domain	R561S	0.03	1.00
	SKM1	SNV	SET domain	Y641C	0	0.09
	OCIAML5	SNV		R685H	0.02	1.00
	P31FUJ	INS	post-SET domain	A731fs	N/A	N/A
SUZ12	KY821	SNV	N-terminal	V68G	0.01	0.00
EED	GDM1	SNV	WD40 repeat	D237E	0.28	0.99
JARID2	NKM1	SNV	EZH1/2-binding domain	D259D	0.67	synonymous mutation
PCL2 (MTF2)	MUTZ8	SNV	Tudor domain	T65T	1	synonymous mutation
	MV411	SNV	Unknown function	Y409C	0.14	0.96
LCOR (PALI1)	KG1	SNV	G9A interaction region	E576K	0	0.00
	P31FUJ	DEL		P588fs	N/A	N/A
	SHI1	SNV	Unknown function	G901A	0	0.00

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
