# Peer review of "Polycomb Alterations in Acute Myeloid Leukaemia: From Structure to Function"

_cancers, 2023, doi:10.3390/cancers15061693_

Round 1
Reviewer 1 Report
well presented review manuscript
can be published in present form
Author Response
We thank the reviewer for going through the manuscript and their comment.
Reviewer 2 Report
In this manuscript Bhattacharyya and Bond reviewed genetic alterations of PRC components in AML. Authors aimed mainly to focus on structural and functional features of PRC2. Although this research potentially could add a reasonable contribution to the field, this manuscript needs some improvements in order to be suitable for publication in this journal. Several issues should be addressed by the Authors:
· Very poor figure quality (Figures 1-4; especially Figures 3 and 4). This could reasonably interfere with reader’s possibility to follow and comprehend.
· Structural features of PRC2 complexes are presented in very high detail, however, functional aspects of PRC2 mutations could be more thoroughly discussed.
· Role of PRC2 in normal hematopoiesis could be discussed in more depth.
· Section “PRC2 genetic interactions in childhood and adult AML” would extremely benefit from appropriate visual summary.
· In the text (lines 201-205) Authors mention Section 3.1 and Section 3.2, however, there are no such sections.
Author Response
- Very poor figure quality (Figures 1-4; especially Figures 3 and 4). This could reasonably interfere with reader’s possibility to follow and comprehend.
We thank the reviewer for highlighting this, alongside their other useful comments and suggestions. We agree that the figure quality was not optimal in the original submission, and we have increased the resolution of all images in this revision. We have also changed the colour of the background in Figure 2, which we believe makes it easier to read. We have further provided links in the text to an interactive and high-resolution version of Figure 4 (page 11, lines 374-376), so that readers can explore these data further.
- Structural features of PRC2 complexes are presented in very high detail, however, functional aspects of PRC2 mutations could be more thoroughly discussed.
We agree. We have now included further details on the predicted effects of PRC2 mutations on function in the text, please see page 6, lines 228-241 and page 7, lines 249-257 of the revised manuscript.
- Role of PRC2 in normal hematopoiesis could be discussed in more depth.
We agree. We have now included further details on the role of PRC2 factors in haematopoiesis, please see page 2, lines 83-96 of the revised manuscript.
- Section “PRC2 genetic interactions in childhood and adult AML” would extremely benefit from appropriate visual summary.
Figure 4 was intended as a visual summary of the currently documented genetic interactions in adult and paediatric cohort in as comprehensive a manner as possible (see page 11, line 354), but we agree that this could be made clearer.
As detailed above, we have now increased the quality of this figure and have provided an interactive link where readers can explore these data directly.
- In the text (lines 201-205) Authors mention Section 3.1 and Section 3.2, however, there are no such sections.
The section numbering was inadvertently excluded during journal formatting, we have now changed the references to sections in the revised manuscript (replaced with section names).

Reviewer 3 Report
It's a well written and understandable paper on a complex issue. The paper properly address an important topic which would deserve important development in the future management of acute leukemias.
Author Response
We thank the reviewer for going through the manuscript and their positive comments.
Round 2
Reviewer 2 Report
Authors have revised the manuscript thoroughly and, in my opinion, it is now acceptable for publication.